# *Lactobacillus fermentum* LF31 Supplementation Reversed Atrophy Fibers in a Model of Myopathy Through the Modulation of IL-6, TNF-α, and Hsp60 Levels Enhancing Muscle Regeneration

**DOI:** 10.3390/nu17091550

**Published:** 2025-04-30

**Authors:** Martina Sausa, Letizia Paladino, Federica Scalia, Francesco Paolo Zummo, Giuseppe Vergilio, Francesca Rappa, Francesco Cappello, Melania Ionelia Gratie, Patrizia Proia, Valentina Di Felice, Antonella Marino Gammazza, Filippo Macaluso, Rosario Barone

**Affiliations:** 1Department of Biomedicine, Neurosciences and Advanced Diagnostics, University of Palermo, 90133 Palermo, Italy; martina.sausa@gmail.com (M.S.); federica.scalia02@unipa.it (F.S.); francescopaolo.zummo01@unipa.it (F.P.Z.); giuseppe.vergilio@unipa.it (G.V.); francesco.cappello@unipa.it (F.C.); melaniaionelia.gratie@unipa.it (M.I.G.); valentina.difelice@unipa.it (V.D.F.); antonella.marinogammazza@unipa.it (A.M.G.); rosario.barone@unipa.it (R.B.); 2Department of Theoretical and Applied Sciences, eCampus University, 22060 Novedrate, Italy; letizia.paladino@uniecampus.it; 3The Institute of Translational Pharmacology, National Research Council of Italy (CNR), 90146 Palermo, Italy; francesca.rappa@unipa.it; 4Department of Psychology, Educational Science and Human Movement, University of Palermo, 90128 Palermo, Italy; patrizia.proia@unipa.it; 5Department of Biological, Chemical and Pharmaceutical Sciences and Technologies (STEBICEF), University of Palermo, 90133 Palermo, Italy

**Keywords:** probiotics, alcohol, skeletal muscle, gut–muscle axis

## Abstract

**Background/Objectives**: Recent studies have highlighted the role of the gut–muscle axis, suggesting that modulation of the gut microbiota may indirectly benefit skeletal muscle. This study aimed to evaluate the effects of *Lactobacillus fermentum* (*L. fermentum*) supplementation in a model of muscle atrophy induced by chronic ethanol (EtOH) intake, focusing on inflammatory and antioxidant mechanisms. **Methods**: Sixty 12-month-old female Balb/c mice were divided randomly into three groups (*n* = 20/group): (1) Ethanol (EtOH) group, receiving ethanol daily for 8 and 12 weeks to induce systemic oxidative stress and inflammation; (2) Ethanol + Probiotic (EtOH + P) group, receiving both ethanol and *L. fermentum* supplementation for the same durations; and (3) Control (Ctrl) group, receiving only water. Muscle samples were analyzed for the fiber morphology, inflammatory markers, oxidative stress indicators, and satellite cell (SC) activity. All data were tested for normality using the Shapiro–Wilk test before applying a parametric analysis. A statistical analysis was performed using one-way ANOVA followed by a Bonferroni post-hoc test. The level of significance was set at *p* < 0.05. **Results**: EtOH exposure caused significant atrophy in all muscle fiber types (type I, IIa, and IIb), with the most pronounced effects on oxidative fibers. *L. fermentum* supplementation significantly reversed atrophy in type I and IIa fibers, accompanied by a significant reduction in IL-6, TNF-α, and Hsp60 expression levels, indicating the protective effect of *L. fermentum* against oxidative stress and inflammation. Moreover, the probiotic treatment increased MyoD expression in SCs, suggesting enhanced regenerative activity, without histological evidence of fibrosis. **Conclusions**: These findings suggest that *L. fermentum* supplementation could counteract EtOH-induced skeletal muscle damage by reducing inflammation and oxidative stress and promoting muscle repair, indicating its potential as an adjuvant, in the therapeutic strategy of models of muscle degeneration.

## 1. Introduction

In recent years, scientific interest in the gut–muscle axis has grown exponentially, parallel to the emergence of new studies that emphasize the key role of the gut microbiota in regulating metabolic, immune, and inflammatory processes [1,2,3]. Alterations in the composition and function of the microbiota, often referred to as dysbiosis, can adversely affect skeletal muscle mass and function, contributing to conditions such as sarcopenia, frailty, and a loss of physical performance, particularly in elderly individuals or in the presence of chronic disease [3,4,5].

Alcohol is a well-recognized toxic agent that adversely affects multiple organ systems, including skeletal muscle. Its capacity to induce gut dysbiosis, increase intestinal permeability, and trigger systemic inflammation makes it a valid experimental model for inducing controlled skeletal muscle damage mediated at least in part by alterations in the gut [6,7,8,9]. Although alcohol is known to have detrimental effects on various organs, the aim of the present study is not to address alcohol-related pathology in a general sense but rather to investigate the emerging role of the gut–muscle axis, a concept that describes the bidirectional interaction between the gut microbiota and muscle function [4,5,10].

Several lines of evidence have indicated that alcohol induces alterations in molecular pathways involved in protein catabolism and anabolism, mitochondrial dysfunction, and the activation of inflammatory processes, all responsible for the onset of sarcopenia [11,12,13,14,15,16]. EtOH consumption leads to the increased production of reactive oxygen species (ROS), exacerbating oxidative stress within skeletal muscle fibers. This oxidative environment damages not only cellular structures but also activates pro-inflammatory signaling pathways, further contributing to alcohol-induced muscle instability [17].

Moreover, mitochondrial impairment caused by chronic alcohol exposure compromises energy metabolism, leading to decreased ATP production, which is essential for proper muscle contraction and endurance [18]. Another critical aspect of alcohol-induced muscle atrophy is its effect on muscle progenitor cells, also known as satellite cells (SCs). These cells play a crucial role in muscle regeneration and repair, yet alcohol consumption significantly reduces their proliferative and differentiation capacity, further contributing to muscle mass loss and impaired muscle growth [19].

There are currently several nutritional strategies aimed at improving muscle mass and/or limiting muscle loss, as well as curbing inflammation and oxidative stress. One of the most widely used approaches is the supplementation of proteins and micronutrients, which, by acting directly on muscle tissue, promote the so-called muscle hypertrophy [20]. However, recent research has shifted attention to alternative nutritional strategies that exploit the gut–muscle axis to indirectly influence muscle health.

Among these, the use of probiotics has gained increasing interest for their ability to modulate the composition of the gut microbiota and restore gut homeostasis [1,21]. Probiotics are live microorganisms, usually bacteria, that, when administered in adequate amounts, benefit the health of the host. The most used probiotic strains belong to the genera Lactobacillus and Bifidobacterium, components of the normal intestinal microbial flora and found in fermented foods, such as yogurt and dietary supplements [22].

Encouraging results on the use of probiotics for muscle health have been reported, offering interesting prospects for ameliorating EtOH-induced damage at the systemic level. Studies have demonstrated that probiotics can mitigate inflammatory processes by modulating immune responses and reducing circulating levels of pro-inflammatory cytokines, such as IL-6 and TNF-α [23]. Additionally, probiotics have been shown to enhance glycogen synthesis, improving muscle energy availability, and increase ATP production, which is fundamental for muscle contraction, endurance, and recovery from oxidative stress [18].

In this context, *Lactobacillus fermentum* (*L. fermentum*) has emerged as a particularly promising probiotic strain due to its potent antioxidant and anti-inflammatory properties. By reinforcing intestinal barrier integrity and reducing systemic inflammation, *L. fermentum* could exert protective effects on skeletal muscle, counteracting EtOH-induced muscle atrophy through modulation of the gut–muscle axis.

Based on these premises, the present study aimed to investigate the efficacy of *L. fermentum* supplementation in mitigating EtOH-induced oxidative stress and muscle atrophy. By assessing key inflammatory markers, oxidative stress indicators, and muscle regeneration factors, we sought to elucidate the molecular mechanisms through which probiotic administration may confer protective effects on skeletal muscle health.

## 2. Materials and Methods

### 2.1. The Model System: Animals and Diets

All experimental procedures involving animals were conducted in compliance with national and European regulations on animal welfare and were approved by the Italian Ministry of Health (authorization number: 1190/2020-PR). In addition, all experiments were conducted at the AteN Center (Advanced Technologies Network Center) at the University of Palermo, formally authorized by the Ministry of Health (Rome, Italy). The study was performed using 60 healthy female mice (BALB/cAnNHsd), 12 months old, obtained from Harlan laboratories S.r.l. (Udine, Italy). The mice had an average weight of 20–24 g at the start of the experiment. Mice were randomly assigned to control and experimental groups, ensuring unbiased distribution across the group [24].

Animals (BALB/cAnNHsd) were house in standard cages under controlled environmental conditions. The room temperature was maintained at 22 ± 2 °C with a constant 12:12 h light–dark cycle, and humidity levels were kept at 50 ± 10 °C. The mice had free access to food (standard diet 4RF21, Mucedola; Settimo Milanese, Milan, Italy) and water, which was the diet for control mice (Ctrl group, *n* = 20). For the experimental mice, 96% ethanol (EtOH) was added (Girolamo Luxardo, Torrella, Padua, Italy) to account for 15% of their total caloric intake. To allow acclimatization to the caloric diet, the animals were initially given a 5% EtOH caloric content for 2 days, which was increased to a 10% EtOH caloric content for 2 days, followed by a 15% EtOH caloric content diet [25,26] for 8 weeks (8EtOH group, *n* = 10) and 12 weeks (12EtOH group, *n* = 10). The standard caloric intake of the mice (14 kcal per day, 3.6 g of their food) was adjusted by diluting EtOH in water to a final volume of 75 μL, which was administered orally daily using a pipette. Two other groups were included: the EtOH and probiotic diet groups, 8EtOH-P and 12EtOH-P (*n* = 10 per group). These mice received the same EtOH regimen as the 8EtOH and 12EtOH groups, but they were also administered the probiotic *L. fermentum* (Bromatech S.r.l., Milan, Italy) at 10^9^ CFUs (colony-forming units) per animal in water, administered orally via pipette every day, with a two-day break on weekends, for the duration of the experiment. The probiotic was administered 30 min after EtOH to assess its potential to mitigate the damaging effects of EtOH on muscles. Food and water intake was recorded daily, and body weight was measured once per week. The diet was controlled by weighing the food to match the daily caloric requirements of the mice. After 8 and 12 weeks, animals were anesthetized using isoflurane (2–4% for induction and 1–2% for maintenance), which is commonly used in animal models to ensure humane and effective anesthesia during procedures. The depth of anesthesia was monitored by assessing the pedal reflex to ensure adequate sedation before proceeding with any interventions. For sacrifice, animals were euthanized via cervical dislocation following deep anesthesia, in accordance with the ethical guidelines and institutional animal care standards. All procedures were approved by the local ethical committee to endure compliance with the ethical treatment of animals in research. Muscle samples were partly frozen in liquid nitrogen and partly dissected and fixed in an acetone:methanol:water solution (2:2:1 ratio) for histological and immunological analysis [27]. The serum was separated via centrifugation at 3000× *g* at 4 °C for 15 min and stored at −80 °C. The sampling process was performed after 48 h after the last treatment with either EtOH alone or EtOH combined with the probiotic, depending on the group.

### 2.2. Histopathology

Muscle samples (gastrocnemius, soleus, and plantaris), initially fixed in a 2:2:1 mixture of acetone, methanol, and water, underwent a graded EtOH dehydration (70, 96, and 100% *v*/*v*). Following this step, the samples were cleared in xylene and embedded in paraffin blocks, as previously described [Barone, 2016 [27]]. Thin sections of 5 μm were obtained from the paraffin blocks and stained with hematoxylin–eosin (E & E) (Appendix A) and by Masson’s trichrome (Bio-Optica, Bergamo, Italy, W01030799) (Appendix A) for a morphological evaluation. Two independent observers evaluated the slides, and images for the morphological analysis were acquired using an Axioscope 5/7 KMAT optical microscope (Carl Zeiss, Milan, Italy) equipped with an Axiocam 208 color digital camera (Carl Zeiss, Milan, Italy).

### 2.3. Immunofluorescence

For the immunofluorescence analysis, muscle sections were incubated in antigen unmasking solution (10 mM trisodium citrate, pH 6.0, Tween-20 0.05%) for 8 min at 75 °C and treated with blocking solution (BSA 3% in PBS) for 30 min at 23 °C. Subsequently, tissues were incubated with the primary antibody (anti-Pax7, anti-MyoD, anti-laminin, or anti-Hsp60), and the sections were incubated in a humidified chamber overnight at 4 °C (Table 1). The following day, the sections were incubated for 1 h at 23 °C with appropriate secondary antibodies (anti-mouse ATTO-488; anti-rabbit ATTO-674) diluted 1:100 in PBS. This was followed by a 15 min incubation at 23 °C with DAPI Fluka (Sigma-Aldrich, St. Louis, MO, USA, 32670) diluted 1:1000 in PBS. Finally, images were captured using a Leica TCS SP8 confocal microscope (Leica Microsystems, Wetzlar, Germany).

### 2.4. Immunohistochemistry

For immunohistochemical analysis, consecutive tissue sections were treated with an antigen retrieval buffer to expose masked epitopes (10 mM trisodium citrate, 0.05% Tween-20, pH 6.0) for 8 min at 90 °C and then immersed in acetone at −20 °C for 5 min to prevent sections from detaching from the sides. All subsequent reactions were conducted at 23 °C. After a wash with PBS (pH 7.4) for 5 min, the sections were immune-stained, using NovolinkPolymer Detection Systems biotin–streptavidin-labeled methodology (RE7140-CE, Leica Biosystems Nussloch, Deer Park, CA, USA). The sections were incubated in primary antibody (Table 1), including anti MHC-I, anti-MHC-IIa, and anti-MHC-IIb, in a humidified chamber overnight at 4 °C. On the next day, they were further incubated for 30 min with post-primary, according to the manufacturer’s instructions. Finally, the slides were cover slipped and evaluated using an Axioscope 5/7 KMA optical microscope (Carl Zeiss, Milan, Italy) equipped with an Axiocam 208 color digital camera (Carl Zeiss, Milan, Italy). The cross-section area (CSA) of type I, type IIa, and type IIb muscle fibers of the posterior hindlimb muscle group were measured using ImageJ 1.41 software to evaluate the effect of treatment with EtOH alone and EtOH plus *L. fermentum*. Analyses were performed using 3 fields per section (40 μm between sections) and 5 mice per group. An average of 352 fibers were analyzed for each mouse.

### 2.5. Enzyme-Linked Immunoadsorbent Assay (ELISA)

The ELISA test was performed on muscle tissue samples frozen at −80 °C. After thawing the muscles and washing them in PBS to remove excess blood, 100 mg of tissue was manually homogenized in 1 mL of 0.02 M PBS, pH 7.2, supplemented with Protease Inhibitor Cocktail (Sigma-Aldrich, St. Louis, MO, USA). The cells were lysed through freezing (−20 °C)/thawing (23 °C) 3 consecutive times. The homogenized tissue was then centrifuged at 5000× *g* for 5 min, and the supernatant was collected for protein analysis. ELISA assays for IL-6, IL-10, TNF-α, and Hsp60 were used following the manufacturer’s instructions (Table 2).

### 2.6. Immunoblotting

Frozen sections of the posterior hindlimb muscles (approximately 50 mg, including gastrocnemius, soleus, and plantaris) were used for skeletal muscle homogenization, as previously described [26]. Subsequently, homogenate samples were centrifuged at 13,000× *g* for 15 min at 4 °C. The resulting supernatants, representing total lysates, were collected and stored at −20 °C. The protein content was measured spectrophotometrically using the Bradford method (Bradford, 1976), with bovine serum albumin (BSA, Sigma-Aldrich) as the calibration standard.

Protein samples were resolved on a 12% gel and subsequently transferred onto a 0.45 μm nitrocellulose membrane (Bio-Rad Laboratories, Segrate Milano, Italy). The membrane was blocked for 1 h at room temperature (23 °C) using Tris-buffered saline (20 mM Tris, 137 mM NaCl, pH 7.6) with 0.05% Tween-20 (T-TBS) and 5% BSA. Following blocking, membranes were incubated with primary antibodies targeting-MyoD, Pax7, or recombinant actin. Primary antibodies were diluted in T-TBS supplemented with 0.5% BSA and incubated with the membranes overnight at 4 °C. The following day, membranes were washed with T-TBS and incubated for 1 h with HRP-conjugated secondary antibodies: goat anti-rabbit IgG (12-348, Sigma-Aldrich, St. Louis, MO, USA) or goat anti-mouse IgG (AB_228297, Invitrogen, Waltham, MA, USA), also diluted in T-TBS with 0.5%. Immunoreactive bands were visualized using the ECL Western Blotting Detection Reagent (Amersham Biosciences, Amersham, UK), following the manufacturer’s instructions. Band intensities were quantified using ImageJ software (version 1.41, NIH, Bethesda, MD, USA; https://imagej.net/ij). Actin served as the internal control for normalization. All blots shown are representative of at least three independent experiments.

### 2.7. Statistical Analysis

All data were collected and analyzed for statistical significance. The mean ± standard deviation (SD) of the mean for each variable outcome was tabulated for each group and for each time-point of the experiments. The data distribution was assessed using the Shapiro–Wilk test to confirm normality. Comparisons among groups and different time points were performed using a One-Way ANOVA test. Subgroup comparisons were carried out using a Bonferroni post hoc test to adjust for multiple comparisons. All statistical analyses were performed using GraphPad Prism^TM^ 4.0 software (GraphPad Software Inc., San Diego, CA, USA). The threshold of statistical significance was set at *p* < 0.05.

## 3. Results

### 3.1. Effects of L. fermentum in Various Fiber Types

Immunohistochemistry analyses of myosin heavy chain (MHC)-I, MHC-IIa, and MHC-IIb (Figure 1) were performed on serial cross-sections to evaluate whether *L. fermentum* had a fibro-specific effect. Neither type IIx nor hybrid fibers were evaluated in this study.

The effects of *L. fermentum* on different muscle fiber types were analyzed after 8 (Figure 2) and 12 (Figure 3) weeks of treatment with EtOH alone or with EtOH + P and compared to the respective Ctrl group. Serial cross-sections of muscles (soleus, plantaris, red gastrocnemius, and white gastrocnemius) analyzed via immunohistochemistry revealed that EtOH treatment for 8 weeks induced the atrophy of type IIa fibers in the red portion of the gastrocnemius muscle (*p* < 0.05) (Figure 2C) and type IIb fibers in the white portion of the gastrocnemius muscle (*p* < 0.05) (Figure 2D) compared to the Ctrl group. No significant changes were observed in type I fibers at this time point. Co-treatment with EtOH + P increased the CSA of type I fibers in the soleus compared to the EtOH-only group (*p* < 0.001) (Figure 2A). A significant increase was also observed in the CSA of type IIa fibers in both the plantaris muscle (*p* < 0.05) (Figure 2B) and the red region of the gastrocnemius (*p* < 0.001) (Figure 2C), compared to the EtOH-only group. Additionally, type IIb fibers in the white region of the gastrocnemius showed an increased CSA in the co-treatment group compared to the EtOH-only group (*p* < 0.001) (Figure 2D). No significant changes were observed in type I fibers of the red gastrocnemius at this stage.

After 12 weeks of treatment, EtOH induced the atrophy of type IIa fibers in the plantaris (*p* < 0.05) (Figure 3B) and the atrophy of type IIb fibers in the white portion of the gastrocnemius (*p* < 0.001) (Figure 3D) compared to the Ctrl group. However, the atrophy of these fibers is counteracted by co-treatment with the *L. fermentum*, both for type IIa fibers in the plantaris (*p* < 0.05) (Figure 3C) and type IIb fibers in the white region of the gastrocnemius (*p* < 0.05) (Figure 3D). Furthermore, *L. fermentum* was able to increase the CSA of type I fibers in the soleus muscle (*p* < 0.001) (Figure 3A) and in the red portion of the gastrocnemius (*p* < 0.05) (Figure 3C) compared to the group treated with EtOH alone. Finally, an increase in the CSA of the type IIa fibers in the red region of the gastrocnemius compared to the EtOH-only group (*p* < 0.05) (Figure 3C) and of the same fibers in the soleus muscle (Figure 3A) was observed not only compared to the EtOH-only group (*p* < 0.0001) but also compared to the Ctrl group (*p* < 0.001). No significant differences were detected in type IIb fibers in the plantaris muscle.

Finally, in order to assess whether *L. fermentum* had a greater effect on the CSA of a specific type of muscle fiber after 8 or 12 weeks, the CSA of muscle fibers at the two treatment times was analyzed. As can be seen from Figure 4, treatment with *L. fermentum* for 12 weeks induced a significant increase in the CSA of both slow (*p* < 0.05) and fast (*p* < 0.001) oxidative fibers in the soleus muscle (Figure 4A) compared to the group of animals treated for 8 weeks. In contrast, a reduction in the CSA of type IIb glycolytic fibers in the white portion of the gastrocnemius muscle was observed in the animals treated for 12 weeks compared to the group of animals treated for 8 weeks (*p* < 0.05) (Figure 4D). CSA values for other types did not show statistically significant differences across time points.

### 3.2. L. fermentum Attenuates Inflammation and Oxidative Stress Induced by Chronic Ethanol Consumption

To determine the possible anti-inflammatory effects of *L. fermentum* administration at the local level, IL-6, IL-10, and TNF-α levels were measured via ELISA, directly in skeletal muscle homogenates, in the two treatment periods (8 and 12 weeks). As shown in Figure 5, EtOH administration for 8 weeks increased IL-6 (*p* < 0.05) (Figure 5A) and IL-10 (*p* < 0.05) (Figure 5B) levels compared to the Ctrl group and reduced TNF-α levels (*p* < 0.001) (Figure 5C) compared to the Ctrl group. On the contrary, co-treatment with *L. fermentum* for 8 weeks reduced both IL-6 levels not only compared to the EtOH-only group (*p* < 0.05) but also compared to the Ctrl group (*p* < 0.001) (Figure 5A) and TNF-α levels compared to the Ctrl group (*p* < 0.001) (Figure 5C). After 12 weeks of treatment, only a change in TNF-α levels was observed, in particular, an increase in this cytokine in the EtOH-treated mice compared to the Ctrl group (*p* < 0.05) and a reduction in TNF-α levels in the animals co-treated with EtOH + P compared to the EtOH-only group (*p* < 0.001) (Figure 5C).

The antioxidant effects of *L. fermentum* were assessed by means of ELISA tests, directly measuring the expression levels of Hsp60, a known mitochondrial marker of oxidative stress, mainly expressed by type I and IIa oxidative fibers compared to type IIb glycolytic fibers, in skeletal muscle, as shown in the immunofluorescence analysis (Figure 6A). The administration of *L. fermentum* for 12 weeks significantly reduced Hsp60 levels not only compared to the EtOH-only group (*p* < 0.0001) but also compared to the Ctrl group (*p* < 0.001) (Figure 6C), demonstrating a possible antioxidant effect.

### 3.3. Administration of L. fermentum Increases the Expression of Pax7 and MyoD in SCs

To evaluate whether *L. fermentum* influenced myogenic progression, the expression levels of Pax7 and MyoD in the skeletal muscle of mice treated for 12 weeks were assessed. Pax7 and MyoD are two nuclear markers (Figure 7A) responsible for the maintenance of the stemness and differentiation of SCs, respectively. Western blotting analysis (Figure 7B) revealed a significant reduction in Pax7 expression levels following EtOH treatment for 12 weeks (*p* < 0.001). In contrast, the administration of *L. fermentum* resulted in the restoration of Pax7 expression levels (*p* < 0.001) and a significant increase in MyoD expression levels (*p* < 0.001) compared to the group of mice treated with EtOH alone.

## 4. Discussion

Skeletal muscle dysfunction can result from various stress-induced conditions, including chronic alcohol consumption, strenuous exercise, aging, cachexia, fibromyalgia, and metabolic disorders [28,29,30,31,32]. These conditions are characterized by a progressive reduction in muscle mass and function, which particularly affects fast-twitch fibers. This impairment is mainly caused by oxidative stress, systemic inflammation, and protein metabolism and can induce significant molecular alterations, leading to muscle degeneration and functional decline.

Probiotics, although having direct effects in modulating the homeostasis of the intestinal muco-microbiotic layer [24,33,34,35,36,37], have been shown to also have indirect effects on various organs including the skeletal muscle by enhancing glycogen synthesis, in turn improving muscle energy availability [38] and increasing ATP production, which is fundamental for muscle contraction, endurance, and recovery from oxidative stress [18]. In this context, *L. fermentum* has emerged as a particularly promising probiotic strain due to its potent antioxidant and anti-inflammatory properties. By reinforcing the muco-microbiotic layer integrity and reducing systemic inflammation, *L. fermentum* could exert protective effects on skeletal muscle, counteracting muscle atrophy through modulation of the gut–muscle axis.

This study aimed to evaluate the potential protective effects of *L. fermentum* supplementation in mitigating skeletal muscle damage induced by chronic EtOH intake in a murine model.

Our results confirmed that prolonged exposure to EtOH induces significant muscle fiber atrophy across different fiber types, particularly type IIa and IIb fibers, with pronounced effects in the gastrocnemius and plantaris muscles after 8 (Figure 2) and 12 weeks of treatment (Figure 3). These findings are in line with previous studies indicating that chronic alcohol consumption impairs muscle protein synthesis, potentially through inhibition of the mTOR signaling pathway, and promotes muscle wasting, especially in muscles rich in glycolytic fibers, such as the gastrocnemius [39]. Conversely, co-treatment with *L. fermentum* significantly preserved the CSA of type I and IIa fibers, suggesting a fiber-type-specific protective effect that may be linked to the metabolic properties of oxidative muscle fibers.

Interestingly, despite the plantaris muscle being predominantly glycolytic with a higher proportion of type IIb fibers [40], *L. fermentum* exerted notable protective effects on the minority population of type IIa fibers. This observation suggests a potential influence of probiotic treatment on mitochondrial function and metabolic adaptation within the muscle, potentially through the activation of signaling pathways involved in mitochondrial biogenesis and oxidative metabolism. Previous studies have demonstrated that probiotic supplementation enhances the mitochondrial content and function in skeletal muscle, thereby reducing oxidative stress and supporting the endurance capacity, particularly through the promotion of oxidative fiber composition and increased mitochondrial enzyme activity [41,42].

Alcohol metabolism is a major contributor to ROS production and the onset of oxidative stress, which, in turn, triggers a cascade of detrimental events including lipid peroxidation, protein oxidation, mitochondrial damage, and the activation of pro-inflammatory pathways [43,44,45]. Consistent with this, we observed elevated levels of Hsp60, a key marker of mitochondrial stress, in the skeletal muscle of mice treated with EtOH for 12 weeks (Figure 6C). Hsp60 plays a dual role in cellular physiology: under normal conditions, it functions as a mitochondrial chaperone, ensuring proper protein folding; under stress conditions, its expression increases as a defense mechanism to prevent protein aggregation and maintain mitochondrial integrity [46,47]. The reduction in Hsp60 expression following *L. fermentum* supplementation suggests that the probiotic may restore the redox balance and enhance mitochondrial resilience. This effect may be partially mediated by the increased availability of endogenous antioxidants, such as glutathione, produced by certain probiotic strains. The improved oxidative status likely reduced the need for elevated Hsp60 expression, thus reflecting a decline in mitochondrial stress. These findings are supported by evidence indicating that Lactobacillus species can reduce ROS production and improve mitochondrial efficiency in various tissues [48].

Notably, Hsp60 is not confined to the mitochondria [49,50,51]. In pathological conditions, it can translocate to the cytoplasm and extracellular space, where it plays immunomodulatory roles. For example, cytoplasmic Hsp60 has been shown to interact with the IκB kinase (IKK) complex, promoting the activation of nuclear factor kappa-light-chain-enhancer of activated B cells (NF-κB), a key transcription factor involved in inflammation and immune responses [52]. In our previous study, we reported increased NF-κB activity in mice exposed to EtOH for 12 weeks, and in this study, we propose a potential link between elevated Hsp60 and NF-κB activation via the VDAC1-dependent release of Hsp60 from mitochondria [53]. Once in the cytoplasm, Hsp60 within the cytoplasm could promote TNF-α-mediated activation of the IKK/NF-κB pathway, demonstrated by increased local TNF-α levels in the gastrocnemius muscle of alcohol-fed mice for 12 weeks (Figure 6C).

Consistent with this hypothesis, our data showed increased levels of TNF-α in EtOH-treated mice, particularly in the gastrocnemius muscle. In parallel, we observed elevated levels of IL-6 and IL-10 after 8 weeks of alcohol administration (Figure 5A,B). IL-6 is a well-known pro-inflammatory cytokine that plays a key role in the acute-phase response, while IL-10 is traditionally considered anti-inflammatory [54]. However, the simultaneous increase in IL-6, TNF-α, and IL-10 suggests a complex and possibly deregulated inflammatory response. While IL-10 generally acts to suppress excessive inflammation and restore tissue homeostasis, recent studies have revealed its pleiotropic nature, showing that it can also exhibit pro-inflammatory effects under certain conditions [4,55,56]. Our findings suggest that in the context of chronic alcohol consumption, IL-10 may fail to adequately control inflammation, thereby contributing to sustained muscle damage. To our knowledge, this is the first study to suggest a possible pro-inflammatory role of IL-10 in skeletal muscle during alcohol-induced oxidative stress, potentially mediated by NF-κB. Importantly, co-treatment with *L. fermentum* reduced IL-6 and TNF-α levels after 8 and 12 weeks, respectively (Figure 5A,C). These findings are in line with previous research indicating that probiotic supplementation with Lactobacillus strains can attenuate systemic inflammation by modulating the gut microbiota composition and reducing the translocation of endotoxins, such as LPSs, from the gut to systemic circulation [57,58,59]. Through these mechanisms, probiotics help to maintain immune homeostasis and reduce the low-grade chronic inflammation typically associated with alcohol abuse and aging.

Despite the presence of elevated inflammatory markers in EtOH-treated mice, we did not observe histological signs of muscle fibrosis (Appendix A), which contrasts with previous reports where collagen deposition and extracellular matrix (ECM) protein expression were up-regulated following alcohol administration [60]. This discrepancy may be attributed to differences in the method of EtOH delivery. While some studies used intraperitoneal injections, which may directly damage muscle tissue and trigger fibrotic responses, our study used oral EtOH administration, which more closely mimics human alcohol consumption patterns and may have a less pronounced fibrogenic effect.

Another significant finding of this study is the effect of *L. fermentum* on muscle regeneration, which is critically dependent on the activity of SCs [61,62]. We found that chronic EtOH consumption suppressed the expression of Pax7 and MyoD, key markers of SC maintenance and myogenic differentiation, respectively (Figure 7B). In contrast, probiotic treatment restored Pax7 expression and significantly enhanced MyoD levels, suggesting that *L. fermentum* supports muscle regeneration by promoting SC activation and differentiation. These findings align with previous reports showing that probiotics can improve muscle function and increase the expression of myogenic factors in models of muscle wasting [63,64].

The role of NF-κB in myogenesis remains controversial. Some studies have shown that NF-κB activity declines during myogenic differentiation and that its inhibition promotes muscle formation. Conversely, other reports suggest that NF-κB can be activated during differentiation, particularly in response to growth factors such as insulin-like growth factor 1 (IGF-1). The dual nature of NF-κB signaling may depend on the context, intensity, and duration of activation [63,65]. In our model, chronic alcohol consumption may have induced sustained NF-κB activation, thereby impairing myogenic signaling (see Figure 8 for a working hypothesis). Our data showed an increase in MyoD-positive SCs following *L. fermentum* supplementation (Figure 7B), supporting the hypothesis that probiotics may help rebalance this pathway, promoting the repair of alcohol-damaged muscle fibers.

Finally, the fiber-type specificity of the protective effects observed with *L. fermentum* supplementation warrants further investigation. Although all fiber types were affected by EtOH, the probiotic preferentially preserved oxidative fibers, which are more dependent on the mitochondrial function and resistance to fatigue. This pattern supports the hypothesis that *L. fermentum* enhances mitochondrial biogenesis and metabolic flexibility, thereby providing a selective advantage to oxidative muscle fibers [41]. Future studies should explore the underlying molecular mechanisms and assess the long-term efficacy of probiotic interventions in different models of muscle degeneration.

## 5. Conclusions

In conclusion, our study provides strong evidence that oral supplementation with *L. fermentum* mitigates the skeletal muscle atrophy induced by EtOH through a combination of anti-inflammatory, antioxidant, and muscle-regeneration mechanisms. These results suggest that probiotics may represent a promising nutritional intervention to preserve muscle function in conditions characterized by chronic oxidative stress and inflammation, such as alcohol myopathy and/or in other models of muscle degeneration. Furthermore, based on its demonstrated protective effects, we propose future investigations on the potential application of this probiotic strain in other in vivo experimental models of muscle deterioration, such as cancer-induced cachexia and fibromyalgia [66,67]. However, future studies should investigate the long-term effects of probiotic supplementation, as well as potential synergistic interactions with other therapeutic strategies targeting the gut–muscle axis.

## Figures and Tables

**Figure 1 nutrients-17-01550-f001:**
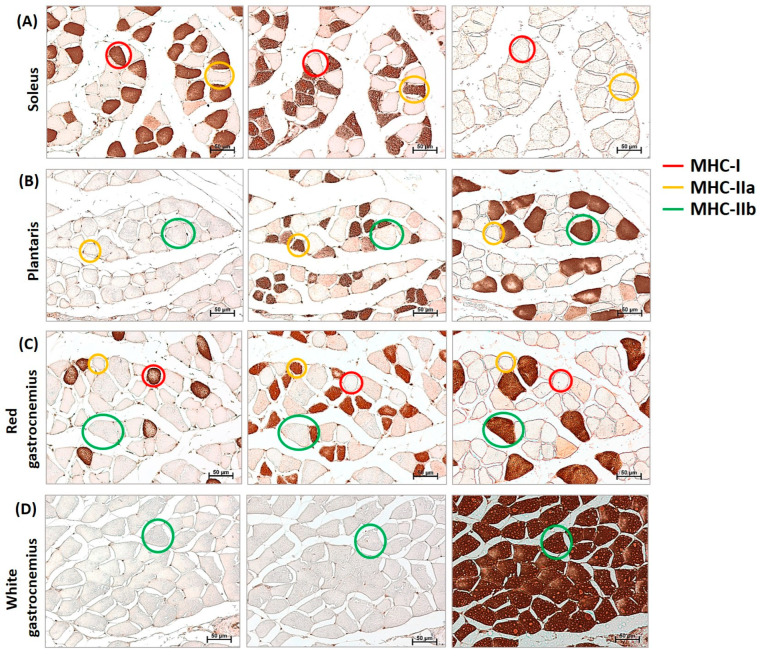
Identification of MHC-I (myosin heavy chain), IIa, and IIb in cross-sections of mouse muscles. Representative example of immunohistochemistry for MHC-I (red circle), MHC-IIa (yellow circle), and MHC-IIb (green circle) in serial cross-sections of the soleus (**A**), plantaris (**B**), red gastrocnemius (**C**), and white gastrocnemius (**D**) muscles of mice. Neither type IIx nor hybrid fibers were evaluated in this study. Images were captured at a magnification of 200×. Bar 50 μm.

**Figure 2 nutrients-17-01550-f002:**
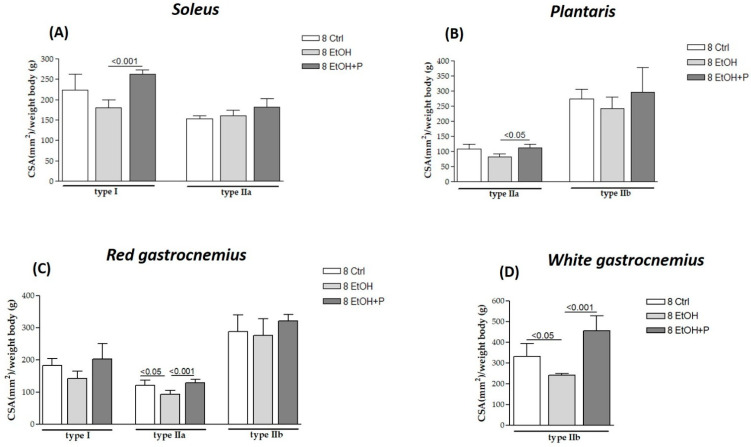
Cross-sectional area (CSA) of type I, IIa, and IIb muscle fibers in the soleus (**A**), plantaris (**B**), red gastrocnemius (**C**), and white gastrocnemius (**D**) muscles in mice treated for 8 weeks with water (8 Ctrl), ethanol (8 EtOH), and ethanol + probiotic (8 EtOH + P), respectively. Analyses were performed using three fields per section (40 μm between sections) and five mice per group. An average of 352 fibers were analyzed for each mouse. The mean of the CSA values obtained was normalized to the body weight of the corresponding mouse [27]. Data are presented as the mean ± SD.

**Figure 3 nutrients-17-01550-f003:**
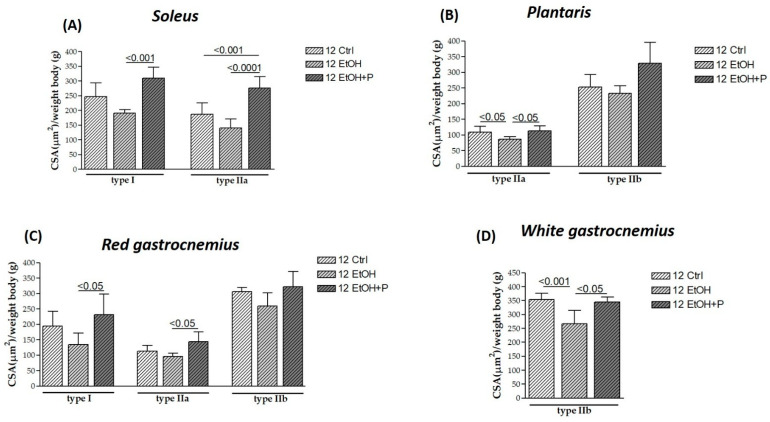
Cross-sectional area (CSA) of type I, IIa, and IIb muscle fibers in the soleus (**A**), plantaris (**B**), red gastrocnemius (**C**), and white gastrocnemius (**D**) muscles in mice treated for 12 weeks with water (12 Ctrl), ethanol (12 EtOH), and ethanol + probiotic (12 EtOH + P), respectively. Analyses were performed using three fields per section (40 μm between sections) and five mice per group. An average of 352 fibers was analyzed for each mouse. The mean of the CSA values obtained was normalized to the body weight of the corresponding mouse [27]. Data are presented as the mean ± SD.

**Figure 4 nutrients-17-01550-f004:**
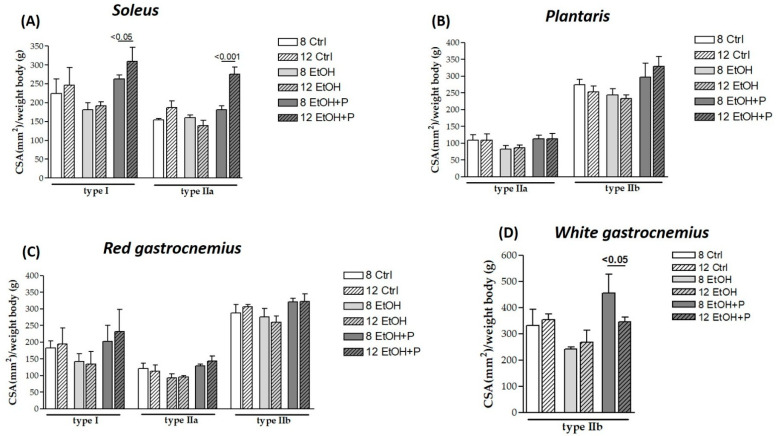
Comparison between the cross-sectional area (CSA) of type I, IIa, and IIb muscle fibers in the soleus (**A**), plantaris (**B**), *red* gastrocnemius (**C**), and white gastrocnemius (**D**) muscles in mice treated for 8 and 12 weeks with water, ethanol, and ethanol + probiotic, respectively. Analyses were performed using three fields per section (40 μm between sections) and five mice per group. An average of 352 fibers was analyzed for each mouse. The mean of the CSA values obtained was normalized to the body weight of the corresponding mouse [27]. Data are presented as mean ± SD.

**Figure 5 nutrients-17-01550-f005:**
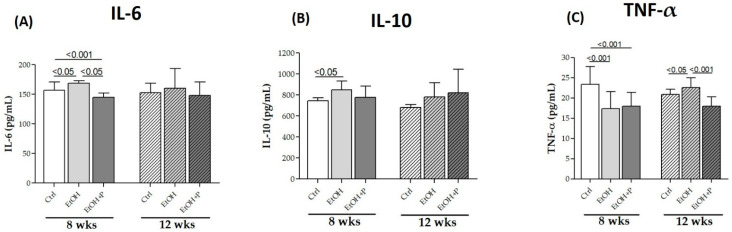
ELISA assays of the cytokines IL-6 (**A**), IL-10 (**B**), and TNF-α (**C**) on whole murine model hindlimb muscle lysates after 8 and 12 weeks of treatment. The analysis was conducted on six mice in each group. Data are presented as the mean ± SD.

**Figure 6 nutrients-17-01550-f006:**
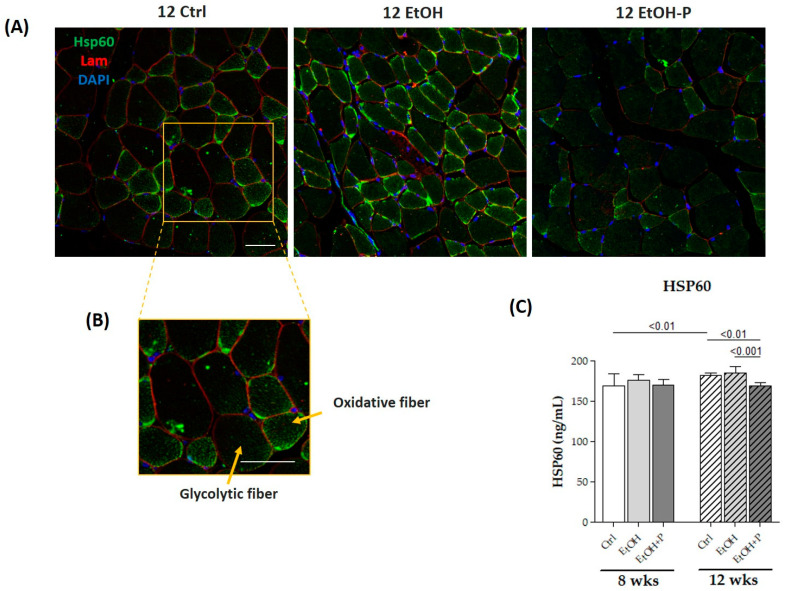
(**A**) Representative immunofluorescence images of Hsp60 tissue distribution in skeletal muscle of mice fed a standard diet (12 Ctrl), ethanol (12 EtOH), and ethanol plus *L. fermentum* LF31 (12 EtOH + P) for 12 weeks. Scale, 25 µm bar. (**B**) Enhanced magnification of the images shown in (**A**). (**C**) ELISA assays of Hsp60 on whole murine model hindlimb muscle lysates after 8 and 12 weeks of treatment. The analysis was conducted on six mice in each group. Data are presented as the mean ± SD.

**Figure 7 nutrients-17-01550-f007:**
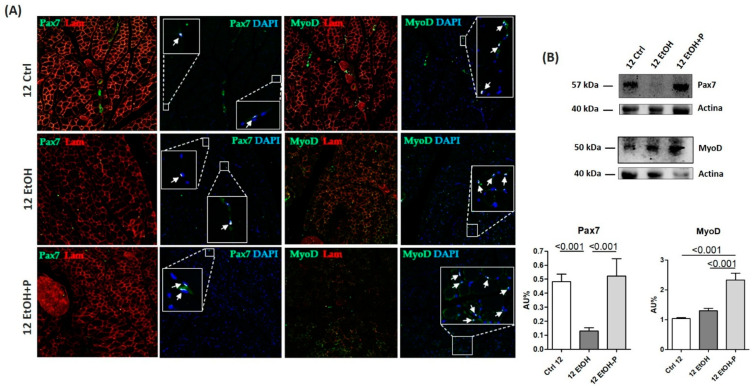
(**A**) Representative immunofluorescence images showing Pax7 and MyoD expression in skeletal muscle sections from mice fed a standard diet (12 Ctrl), ethanol (12 EtOH), and ethanol combined with *L. fermentum* LF31 (12 EtOH + P) for 12 weeks. Arrows indicate colocalization of MyoD and Pax7 with DAPI, indicating nuclear localization of MyoD and Pax7 factors in satellite cells (SCs). Scale, 25 µm bar. (**B**) Representative Western blots of Pax7 (57 kDa) and MyoD (50 kDa) in skeletal muscle tissues from mice (*n* = 3) fed a standard diet (12 Ctrl), ethanol (12 EtOH), and ethanol plus *L. fermentum* LF31 (12 EtOH + P) for 12 weeks; 40 µg of protein was loaded in each lane; actin (40 kDa) was used as the loading control. Data are presented as the mean ± SD.

**Figure 8 nutrients-17-01550-f008:**
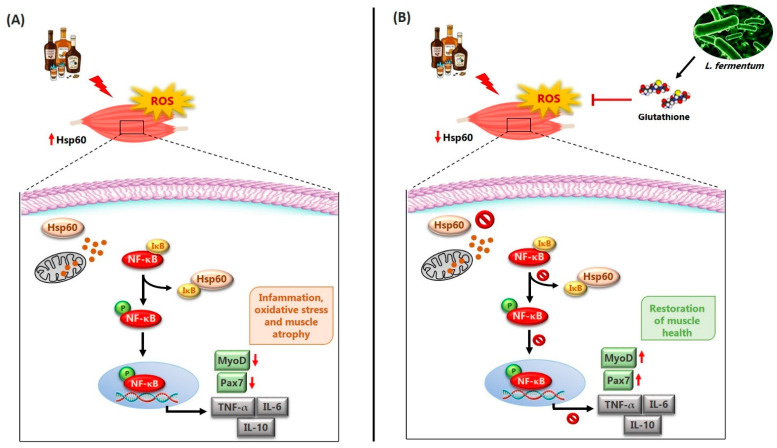
Working hypothesis. (**A**) Chronic alcohol consumption leads to an increase in circulating reactive oxygen species (ROS) and, locally, to an increase in the mitochondrial protein Hsp60. Hsp60, in the presence of oxidative stress, could translocate from the mitochondria to the cytoplasm via a voltage-dependent anion channel located on the mitochondrial membrane and interact directly with IκB. The dissociation of IκB from NF-κB would induce the phosphorylation of NF-κB (pNF-κB) and its translocation to the nuclear level, where it could induce the expression of inflammatory cytokines, including TNF-α, IL-6, and IL-10, as well as the reduction of the expression of markers responsible for mechanisms of muscle regeneration and differentiation, such as Pax7 and MyoD. These events would be responsible for the atrophy of muscle fibers found in animals treated with alcohol for 8 and 12 weeks. (**B**) The potential effects of *L. fermentum* may involve modulation of the NF-κB signaling pathway [24]. The probiotic *L. fermentum*, through the production of the antioxidant glutathione, could induce the reduction of local Hsp60 levels as a protective mechanism against oxidative stress. In this way, Hsp60 would not be exported from the mitochondria to the cytoplasm and would not interact with IκB. NF-κB would therefore remain associated with IκB in an inactive complex and could not activate the expression of inflammatory cytokines. Consequently, the negative effects of alcohol-induced damage, such as the reduction of markers of muscle regeneration and differentiation (Pax7 and MyoD) at the muscle level, may be enhanced. These phenomena would contribute to counteract the atrophy of skeletal muscle fibers induced by chronic EtOH consumption, suggesting a possible protective role of the probiotic *L. fermentum*.

**Table 1 nutrients-17-01550-t001:** List of the antibodies used for analysis of immunohistochemistry (IHC), immunofluorescence (IF), and Western blotting (WB).

Primary Antibody	Features	IHC	IF	WB
Paired box 7 (Pax7)	5081-MSM1-P1ABX Mouse Monoclonal NeoBiotecnhologies, Union City, CA, USA	-	1:200	1:1000
Myoblast determination protein 1 (MyoD)	MA1-41017 Mouse Monoclonal Invitrogen, Waltham, MA, USA	-	1:300	1:1000
Actin Recombinant (JJ09-29)	MA5-32479 Rabbit Monoclonal Invitrogen, Waltham, MA, USA	-	-	1:1000
Laminin	AB2034 Rabbit Polyclonal Millipore, Chicago, IL, USA	-	1:50	-
Heat Shock Protein 60 (Hsp60)	Sc-1722 Goat Polyclonal Santa Cruz, CA, USA	-	1:50	-
Type I Myosin Heavy Chain (MHC-I)	A4.951-s Mouse Monoclonal DSHB, Lowa City, IA, USA	1:50	-	
Type IIa Myosin Heavy Chain (MHC-IIa)	SC-71-s Mouse Monoclonal DSHB, Lowa City, IA, USA	1:50	-	
Type IIb Myosin Heavy Chain (MHC-IIb)	BF-F3-s Mouse Monoclonal DSHB, Lowa City, IA, USA	1:50	-	

**Table 2 nutrients-17-01550-t002:** List of the ELISA tests used for analysis their respective sample dilutions and limits of detection.

Target	Features	Sample Dilution	Limit of Detection	Precision
Mouse IL-6	LS-F263 LS Bio, Lynnwood, WA, USA	No	0.82–600 pg/mL	Intra-Assay: CV < 10% Inter-Assay: CV < 12%
Mouse IL-10	LS-F253 LS Bio, Lynnwood, WA, USA	No	45–5000 pg/mL	Intra-Assay: CV < 10% Inter-Assay: CV < 12%
Mouse HSPD1	LS-F4239 LS Bio, Lynnwood, WA, USA	No	4.56–100 ng/mL	Intra-Assay: CV < 10% Inter-Assay: CV < 12%
Mouse TNF alpha	LS-F5192 LS Bio, Lynnwood, WA, USA	No	15.6–1000 pg/mL	Intra-Assay: CV < 10% Inter-Assay: CV < 12%

## Data Availability

Data are contained within the article and Appendix A.

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
