# Peer review of "Lactobacillus fermentum LF31 Supplementation Reversed Atrophy Fibers in a Model of Myopathy Through the Modulation of IL-6, TNF-α, and Hsp60 Levels Enhancing Muscle Regeneration"

_nutrients, 2025, doi:10.3390/nu17091550_

Round 1
Reviewer 1 Report
Comments and Suggestions for Authors
The aim of this study was to examine the effect of probiotic, Lactobacillus fermentum, supplementation on ethanol-induced injury of the skeletal muscles. The results show that Lactobacillus attenuated ethanol-induced atrophy, inflammation and oxidative stress. Ethanol reduced the expression of satellite stem cell markers, Pax7 and MyoD, whereas Lactobacillus fermentum had the opposite effect. Overall, the results indicate that this probiotic could be useful in improving skeletal muscle phenotype in alcohol abusers.
The topic and the results are of some interest, however, there are also important concerns to be addressed.
- All experiments were performed in female mice only. Do you expect the similar results in males?
- The experiments were performed in 12-month old mice; could age-associated sarcopenia contribute to the results obtained?
- The method of animal anesthesia and sacrifice should be described.
- The intra- and inter-assay CVs for ELISA kits used should be specified.
- Statistical analysis, were all data normally distributed to justify using parametric tests only?
- Section 3.1, the effect of both ethanol and probiotic on all types of fibers should be described, even if no significant changes were observed.
- 5, the differences between cytokine levels in the specific group, although significant, were relatively small. Are these differences relevant for the pathogenesis of muscle injury? Why cytokines were expressed per mL rather than per unit of muscle protein?
- How could be interpreted that ethanol increased the anti-inflammatory IL-10 and reduced pro-inflammatory TNF-alpha?
- Was there any effect of Lactobacillus on ethanol pharmacokinetics?
- Are there any data about the effect of ethanol on composition of gut microbiota?
Author Response
Response to Reviewers
We sincerely thank the reviewers and the editor for their time, insightful comments, and constructive suggestions, which have greatly contributed to the improvement of our manuscript.
We would also like to mention that, shortly after the initial submission, we contacted the editorial office to request the possibility of re-submitting the manuscript, as we realized that some additional corrections and clarifications were necessary. We appreciate the opportunity to revise and resubmit.
Following a more in-depth analysis of the literature and in light of the reviewer feedback, we have made further adjustments to the manuscript to enhance clarity, precision, and the overall scientific message. These revisions include rephrasing certain sections, modifying the title, and adjusting parts of the discussion and conclusion to better reflect the broader implications of our findings.
All modifications are clearly indicated in the revised manuscript using track changes. Please find below our point-by-point responses to the reviewers’ comments.
REVIEWER 1
Comment 1: All experiments were performed in female mice only. Do you expect the similar results in males?
Response 1: We appreciate the reviewer's insightful comment. The choice to use female mice was based on both practical and biological considerations. From a methodological point of view, female mice tend to show less aggressive behavior in groups, reducing stress-related variability that could confound muscle physiology and inflammatory responses. Furthermore, literature data suggest that, in humans, females develop alcohol use disorders and related complications (including musculoskeletal disorders) more rapidly than males, due to differences in body composition and alcohol metabolism [1]. These findings support the translational relevance of using females in this preclinical model. However, we fully recognize that sex-specific differences may influence both the pathophysiological response to ethanol and the therapeutic effect of L. fermentum. Future studies will be needed to assess whether these findings extend to males and to explore potential sex-related variations in the gut-muscle axis.
- Shogren, M.D.; Harsell, C.; Heitkamp, T. Screening Women for At-Risk Alcohol Use: An Introduction to Screening, Brief Intervention, and Referral to Treatment (SBIRT) in Women's Health. JMWH 2017, 62(6), 746-754.
Comment 2: The experiments were performed in 12-month old mice; could age-associated sarcopenia contribute to the results obtained?
Response 2: We thank the reviewer for raising this important point. The 12-month-old mice used in our study are considered to be in the early stages of aging, and indeed, age-associated sarcopenia could contribute to the observed muscle atrophy, particularly in the oxidative fibers. While sarcopenia is a well-established consequence of aging, our study focused on the specific impact of chronic ethanol consumption and the potential protective effects of L. fermentum supplementation in mitigating muscle damage induced by oxidative stress and inflammation.
However, we recognize that sarcopenia may have influenced baseline muscle function and response to ethanol exposure in these animals. To address this potential confounding factor, it would be valuable to include younger mice in future studies to determine if the effects of L. fermentum are more pronounced in the absence of age-related muscle degeneration. Furthermore, we plan to investigate the impact of L. fermentum in aged mice with more advanced sarcopenia to better understand its therapeutic potential in age-related muscle degeneration.
Comment 3: The method of animal anesthesia and sacrifice should be described.
Response 3: We appreciate the reviewer’s suggestion. We agree that a more detailed description of the animal sacrifice methods is necessary for clarity and transparency. In our study, animals were euthanized via cervical dislocation, in accordance with ethical guidelines and institutional animal care standards. All procedures were approved by the local ethical committee to ensure compliance with the ethical treatment of animals in research. We included these methodological details in the revised manuscript to provide greater clarity.
Comment 4: The intra- and inter-assay CVs for ELISA kits used should be specified.
Response 4: We thank the reviewer for this valuable suggestion. We recognize the importance of providing detailed information on the precision and reliability of the assays used. The intra- and inter-assay coefficient of variation CVs for the ELISA kits used in this study have been included in the revised manuscript in Table 2. For the specific kits used in our study, the intra-assay CVs were <10% and the inter-assay CVs were <12%, as reported by the manufacturer. These values demonstrate the reliability and consistency of the assays for measuring inflammatory markers and indicators of oxidative stress in muscle tissue samples.
Comment 5: Statistical analysis, were all data normally distributed to justify using parametric tests only?
Response 5: Thank you for your insightful comment. We confirm that the assumption of normality was assessed before applying parametric statistical tests. Specifically, the Shapiro-Wilk test was used to evaluate the normal distribution of the data. Only variables showing a normal distribution were analyzed using parametric tests (One-Way ANOVA followed by Bonferroni post hoc test). This information has now been included in the revised version of the manuscript in the Statistical Analysis section.
Comment 6: Section 3.1, the effect of both ethanol and probiotic on all types of fibers should be described, even if no significant changes were observed.
Response 6: We thank the reviewer for this insightful suggestion. We agree that reporting the effects on all muscle fiber types, even when no significant differences were observed, enhances the clarity and completeness of the results. Accordingly, we have revised Section 3.1 to include a comprehensive description of all fiber types (MHC-I, MHC-IIa, and MHC-IIb) analyzed in each muscle group, specifying when no significant changes were detected in response to ethanol or L. fermentum supplementation. This allows the reader to better interpret the selective or generalized effects of the treatments. The revised text is now included in Section 3.1 of the manuscript.
Comment 7: 5, the differences between cytokine levels in the specific group, although significant, were relatively small. Are these differences relevant for the pathogenesis of muscle injury? Why cytokines were expressed per mL rather than per unit of muscle protein?
Response 7: Regarding the magnitude of cytokine level changes, we acknowledge that the observed differences were moderate in absolute terms. However, even subtle alterations in pro-inflammatory cytokines such as IL-6 and TNF-α are known to have a meaningful biological impact on muscle homeostasis, particularly under chronic exposure conditions such as ethanol-induced stress. These cytokines can disrupt muscle protein synthesis, promote proteolysis, and impair SCs function, contributing to muscle atrophy even at relatively low concentrations, as previously reported in both experimental and clinical settings [2,3]. Therefore, we believe that the changes observed in our model are biologically relevant in the context of early inflammatory dysregulation and muscle degeneration.
As for the expression of cytokine levels in pg/mL, these measurements were obtained from homogenized muscle tissue supernatants following standard extraction protocols, and the results were expressed in pg/mL as per the manufacturer’s ELISA guidelines. We agree that normalization to total muscle protein content could provide additional context; unfortunately, due to technical constraints during sample preparation, total protein concentration was not systematically measured in all samples. This is a limitation of the current study.
- Kumar, A.; Davuluri, G.; Welch, N.; Kim, A.; Gangadhariah, M.; Allawy, A.; Priyadarshini, A.; McMullen, M.R.; Sandlers, Y.; Willard, B.; Hoppel, C.L.; Nagy, L.E.; Dasarathy, S. Oxidative stress mediates ethanol-induced skeletal muscle mitochondrial dysfunction and dysregulated protein synthesis and autophagy. Free Radic Biol Med 2019, 145, 284-299.
- Zhang, X.; Tachibana, S.; Wang, H.; Hisada, M.; Williams, G.M.; Gao, B.; Sun, Z. Interleukin-6 is an important mediator for mitochondrial DNA repair after alcoholic liver injury in mice. Hepatology 2010, 52(6), 2137-47.
Comment 8: How could be interpreted that ethanol increased the anti-inflammatory IL-10 and reduced pro-inflammatory TNF-alpha?
Response 8: We thank the reviewer for pointing out this important observation. The effects of ethanol on immune modulation are complex and context-dependent. Ethanol is generally considered to be pro-inflammatory, particularly in the context of tissue injury and chronic inflammation. However, it is well-documented that ethanol, especially in chronic or low-to-moderate doses, can trigger an adaptive immune response, which may involve both pro- and anti-inflammatory cytokines.
One possible explanation for the increased levels of IL-10 in the presence of ethanol is that the body attempts to counteract the heightened pro-inflammatory response induced by ethanol exposure. IL-10 plays a key role in regulating and dampening inflammation by inhibiting the production of pro-inflammatory cytokines like TNF-α, thus contributing to tissue protection and homeostasis. Therefore, the observed increase in IL-10 might represent a compensatory mechanism by which the immune system tries to control the excessive inflammation and tissue damage induced by ethanol.
Additionally, ethanol exposure may affect the balance between different immune cell populations (e.g., macrophages and T cells), which could shift the response towards a more regulatory profile. It is also possible that IL-10 expression might be up-regulated early in the inflammatory process as part of a protective response to mitigate ethanol-induced damage, which could explain the reduction of TNF-α levels in some conditions.
Further investigations on the specific immune pathways modulated by ethanol in skeletal muscle could help clarify the relationship between these opposing cytokines and provide deeper insights into the role of IL-10 in ethanol-induced muscle injury.
Comment 9: Was there any effect of Lactobacillus on ethanol pharmacokinetics?
Response 9: Thank you for this insightful comment. Our study primarily focused on investigating the effects of L. fermentum supplementation on muscle atrophy, inflammation, and oxidative stress induced by chronic ethanol exposure, rather than on the pharmacokinetics of ethanol itself. Therefore, we did not assess the impact of L. fermentum on ethanol absorption, distribution, metabolism, or excretion.
While gut microbiota has been shown to influence drug metabolism and alcohol absorption in certain studies, L. fermentum supplementation was not specifically tested for any potential modulation of ethanol pharmacokinetics in this experiment. We acknowledge that this could be an important aspect to investigate in future studies, as the gut microbiota is known to play a role in the metabolism of various substances, including alcohol. However, the current study focused on the muscle-related effects and did not aim to assess pharmacokinetic parameters.
Comment 10: Are there any data about the effect of ethanol on composition of gut microbiota?
Response 10: Thank you for your comment. Yes, it is well-documented that chronic ethanol consumption can significantly alter the composition of the gut microbiota, leading to dysbiosis. Ethanol typically reduces microbial diversity and promotes an overgrowth of pathogenic bacteria like Proteobacteria, which can disrupt gut barrier integrity and cause systemic inflammation. This dysbiosis can contribute to various alcohol-related diseases, including liver damage, muscle wasting, and neuroinflammation.
In our study, while we did not directly assess the impact of ethanol on microbiota composition, we hypothesize that the L. fermentum intervention could help modulate gut microbiota and thereby reduce the inflammation and oxidative stress induced by ethanol. L. fermentum has been shown to exert protective effects against ethanol -induced damage through its anti-inflammatory properties, and recent studies, such as the work by Paladino et al. [4], support this hypothesis. Specifically, Paladino et al. demonstrated that L. fermentum supplementation improved the recovery of both the small intestine and cerebellum in mice subjected to ethanol-induced damage by regulating NF-kB signaling and enhancing the function of the chaperone system. This study suggests that the restorative effects of L. fermentum are partly due to its ability to modulate the immune response and potentially mitigate the systemic inflammation caused by alcohol.
Based on this, it is likely that L. fermentum could also help restore gut microbiota balance, improving gut permeability and decreasing the translocation of endotoxins like LPSs, which could help alleviate some of the inflammation and muscle damage observed in our study. However, a more detailed investigation into the microbiota composition and the direct effects of L. fermentum on ethanol pharmacokinetics would be necessary for a more comprehensive understanding of the mechanisms at play.
- Paladino, L.; Rappa, F.; Barone, R.; Macaluso, F.; Zummo, F.P.; David, S.; Szychlinska, M.A.; Bucchieri, F.; Conway de Macario, E.; Macario, A.J.L.; Cappello, F.; Marino Gammazza, A. NF-kB Regulation and the Chaperone System Mediate Restorative Effects of the Probiotic Lactobacillus fermentum LF31 in the Small Intestine and Cerebellum of Mice with Ethanol-Induced Damage. Biology (Basel) 2023, 1, 1394.

Reviewer 2 Report
Comments and Suggestions for Authors
Lactobacillus fermentum LF31 supplementation reversed atrophy fibers in a model of Alcohol-Induced Myopathy through the modulation of IL-6, TNF-α and Hsp60 levels enhancing muscle regeneration
The manuscript examined the role of the gut-muscle axis, suggesting that modulation of the gut microbiota may indirectly benefit skeletal muscle. This study aimed to evaluate the effects of Lactobacillus fermentum (L. fermentum) supplementation on muscle atrophy induced by chronic ethanol (EtOH) intake, focusing on inflammatory and antioxidant mechanisms. Female Balb/c mice were divided into three groups and received one on three treatments. Muscle samples were analyzed for fiber morphology, inflammatory markers, oxidative stress indicators, and satellite cell (SCs) activity. The results revealed that EtOH exposure caused significant atrophy in all muscle fiber types (type I, IIa, and IIb), with the most pronounced effects on oxidative fibers. L. fermentum supplementation significantly reversed atrophy in type I and IIa fibers, accompanied by a significant reduction in IL-6, TNF-α and Hsp60 expression level, indicating the protective effect of L. fermentum against oxidative stress and inflammation. The authors concluded that These findings suggest that L. fermentum supplementation could counteract EtOH-induced skeletal muscle damage by reducing inflammation and oxidative stress and promoting muscle repair, indicating its potential as adjuvant - in the therapeutic strategy in models of muscle degeneration.
L26-29: the treatments should be clarified in the abstract.
All abbreviations should be clarified in the abstract.
Statement of experimental design, statistical analysis, and P values should be added to the abstract.
L50: revise
The introduction is good; however, you may consider changing the paragraph structure.
L116: provide the average weight on the mice used. And how the randomization process took place.
Provide the chemical composition of the diets.
Provide more details about the experimental design, housing, environment.
Provide more details about the process of mice sampling. How many per group?
L222: could you prove the SEM instead of SD
The materials and method section are well prepared however it lacks some information listed above.
Result and discussion sections are well prepared; all tables and graphs are clear and serve the purpose.
Author Response
Response to Reviewers
We sincerely thank the reviewers and the editor for their time, insightful comments, and constructive suggestions, which have greatly contributed to the improvement of our manuscript.
We would also like to mention that, shortly after the initial submission, we contacted the editorial office to request the possibility of re-submitting the manuscript, as we realized that some additional corrections and clarifications were necessary. We appreciate the opportunity to revise and resubmit.
Following a more in-depth analysis of the literature and in light of the reviewer feedback, we have made further adjustments to the manuscript to enhance clarity, precision, and the overall scientific message. These revisions include rephrasing certain sections, modifying the title, and adjusting parts of the discussion and conclusion to better reflect the broader implications of our findings.
All modifications are clearly indicated in the revised manuscript using track changes. Please find below our point-by-point responses to the reviewers’ comments.
REVIEWER 2
Comment 1: L26-29: the treatments should be clarified in the abstract.
Response 1: Thank you for your valuable comment. We have revised the abstract to better clarify the treatments used in the study. Specifically, we have now explicitly mentioned the different experimental groups and the treatments they received, including the ethanol (EtOH) and L. fermentum supplementation (EtOH + Probiotic) groups, as well as the control (Ctrl) group that received only water. We hope this revision makes the experimental design and treatments clearer for the readers.
Comment 2: All abbreviations should be clarified in the abstract.
Response 2: Thank you for your suggestion. We have revised the abstract to clarify all abbreviations used in the text. Each abbreviation is now explained the first time it is mentioned, ensuring better clarify for the readers.
Comment 3: Statement of experimental design, statistical analysis, and P values should be added to the abstract.
Response 3: Thank you for your valuable comment. We have revised the abstract to include a brief description of the experimental design, the statistical analysis performed (using one-way ANOVA followed by Bonferroni post-hoc test), and the P values for statistical significance (set at p<0.05). We hope this revision satisfies your request and enhances the clarity of the study methodology.
Comment 5: L50: revise
Response 5: Thank you for your suggestion. The sentence in line 50 has been revised to improve clarity and scientific accuracy.
Comment 6: The introduction is good; however, you may consider changing the paragraph structure.
Response 6: Thank you for your suggestion. We have restructured the introduction as suggested, aiming to make the paragraphs clearer and more defined, in order to improve the flow and comprehensibility of the text. We have separated the information regarding the gut-muscle axis, the effects of alcohol on muscle health, the role of probiotics, and treatment prospects in a more logical and fluid manner. We believe this new structure helps to better contextualize our study and presents the key concepts in a more coherent way.
Comment 7: L116: provide the average weight on the mice used. And how the randomization process took place.
Response 7: Thank you for your suggestion. We have added the information regarding the average weight of the mice used in the study in paragraph 2.1.
Comment 8: Provide the chemical composition of the diets.
Response 8: Thank you for your valuable comment. The mice in our study were fed a standard diet (4RF21, Mucedola; Settimo Milanese, Milan, Italy). This diet is a commercially available rodent feed. The detailed composition of the 4RF21 diet is available in the product data sheet provided by the manufacturer, and we ensured that the diet met the nutritional needs of the animals throughout the study.
Comment 9: Provide more details about the experimental design, housing, environment.
Response 9: Thank you for your suggestion. We have revised the manuscript to include more detailed information regarding the experimental design, housing, and environmental conditions.
Comment 10: Provide more details about the process of mice sampling. How many per group?
Response 10: Thank you for your comment and suggestion. We have updated the manuscript to provide additional details regarding the sampling process and experimental time frame, as requested. While the number of animals per group is already provided in the manuscript.
Comment 11: L222: could you prove the SEM instead of SD
Response 11: Thank you for your comment. We acknowledge the suggestion to present the data using the SEM instead of SD. However, we chose to use SD in order to better reflect the variability within each experimental group, which we believe is more appropriate given the aim of our study to describe biological differences among groups. Nevertheless, we are open to revising the figures and tables to present SEM instead of SD if the Editorial Board considers this more suitable for the journal's standards.
Comment 12: The materials and method section are well prepared however it lacks some information listed above.
Response 12: Thank you for your comment. We appreciate your feedback on the Materials and Methods section. As mentioned in the previous responses, we have provided the additional information regarding the experimental design, sampling process, and other details that were requested (such as the number of animals per group, sampling method, and experimental time frame). These updates have been incorporated into the revised manuscript. Please let us know if any further clarification is needed.
